# Staged Joint Arthrodesis in the Treatment of Severe Septic Ankle Arthritis Sequelae: A Case Report

**DOI:** 10.3390/ijerph182312473

**Published:** 2021-11-26

**Authors:** Yong-Cheol Hong, Ki-Jin Jung, Hee-Jun Chang, Eui-Dong Yeo, Hong-Seop Lee, Sung-Hun Won, Jae-Young Ji, Dhong-Won Lee, Ik-Dong Yoo, Sung-Joon Yoon, Woo-Jong Kim

**Affiliations:** 1Department of Orthopaedic Surgery, Soonchunhyang University Hospital Cheonan, Cheonan 31151, Korea; ryanhong90@gmail.com (Y.-C.H.); c89546@schmc.ac.kr (K.-J.J.); spixnl1@naver.com (H.-J.C.); yunsj0103@naver.com (S.-J.Y.); 2Department of Orthopaedic Surgery, Veterans Health Service Medical Center, Seoul 05368, Korea; angel_doctor@naver.com; 3Department of Foot and Ankle Surgery, Nowon Eulji Medical Center, Eulji University, Seoul 01830, Korea; sup4036@naver.com; 4Department of Orthopaedic Surgery, Soonchunhyang University Hospital Seoul, Seoul 04401, Korea; orthowon@schmc.ac.kr; 5Department of Anesthesiology and Pain Medicine, Soonchunhyang University Hospital Cheonan, Cheonan 31151, Korea; phmjjy@naver.com; 6Department of Orthopaedic Surgery, Konkuk University Medical Center, Seoul 05030, Korea; wonbayo@naver.com; 7Department of Nuclear Medicine, Soonchunhyang University Hospital Cheonan, Cheonan 31151, Korea; 92132@schmc.ac.kr

**Keywords:** septic arthritis, iliac graft, internal fixation, external fixation

## Abstract

Septic ankle arthritis is a devastating clinical entity with high risks of morbidity and mortality. Prompt treatment is necessary because delayed or inadequate treatment can lead to irreversible damage that may occur on the articular surface, resulting in cartilage erosion, infective synovitis, osteomyelitis, joint deformity, and pain and joint dysfunction. An aggressive surgical approach is required when a joint infection causes severe limb-threatening arthritis. A 58-year-old woman visited our clinic with increasing pain in the right ankle, which had been present for the previous 2 months. She complained of discomfort in daily life due to deformity of the ankle; limping; and severe pain in the ankle even after walking a little. The patient reported a history of right-ankle injury while exiting a bus in her early 20s. Plain radiographs of the right ankle joint revealed that the medial malleolus was nearly absent in the right ankle joint on the anteroposterior view, and severe varus deformity was observed with osteoarthritic changes because of joint space destruction. Magnetic resonance imaging revealed diffuse synovial thickening of the destroyed tibiotalar joint with joint effusion. Hybrid 99mTc white blood cell single-photon emission computed tomography/computed tomography showed increased uptake along the soft tissue around the ankle joint; uptake was generally low in the talocrural and subtalar joints. A two-stage operation was performed to remove the infected lesions and correct the deformity, thus enabling limb salvage. The patient was nearly asymptomatic at the 6-month follow-up, with no discomfort in her daily life and nearly normal ability to carry out full functional activities. She had no complications or recurrent symptoms at the 1-year follow-up. We have described a rare case of a staged limb salvage procedure in a patient with chronic septic arthritis sequelae. For patients with severe joint deformity because of septic ankle sequelae, staged arthrodesis is a reliable method to remove infected lesions, solve soft tissue problems, correct deformities, and maintain leg length.

## 1. Introduction

Septic ankle arthritis is generally rare. Among all patients with infectious arthritis, the prevalence of infectious ankle arthritis is limited (3.4–15%) [1,2,3,4,5]. Because this disease is a devastating clinical entity with high risks of morbidity and mortality, an accurate understanding of the disease, diagnosis, and prompt treatment is essential [6,7].

Infections can be introduced into the joint through hematogenous spread from other infection sites, by direct inoculation due to trauma, or by contiguous spread from adjacent tissues [6]. Risk factors in patients susceptible to septic arthritis include nicotine abuse, obesity, age > 80 years, diabetes, alcoholism, trauma, previous joint surgery, rheumatoid arthritis or osteoarthritis, other inflammatory arthropathies, chronic hypoxia, malignant disease, cutaneous ulcer, hepatic or renal failure, or intravenous or intraarticular injection history [6,7,8,9].

Streptococcus aureus is the most common organism causing infectious joints of the ankle, and early empirical antibiotic treatment is recommended [7,8,9,10,11]. The diagnosis of septic arthritis relies on a modification of the criteria used by Newman [1], which requires one of four factors to be met: (1) isolation of a pathogenic organism from an affected joint; (2) isolation of a pathogenic organism from another source (e.g., blood) in the context of a warm, red joint that is presumed to exhibit sepsis; (3) typical clinical features and a turbid joint in the presence of previous antibiotic treatment; or (4) postmortem or pathological features that are suggestive of septic arthritis.

Destruction of the joint due to septic arthritis results in functional instability and changes in lower limb muscle activity patterns during unexpected disturbances, particularly in proximal muscles rather than distal muscles [12]. In addition, according to a study by Casado-Hernández et al. [13], patients with anterior talofibular ligament (ATFL) injuries showed a greater presence of calcaneofibular ligament and tibiotalar joint injuries than subjects with non-injured. These constitute the rationale that functional instability can lead to more severe joint destruction, which accelerates the patient’s limping and pain and intensifies discomfort.

The principles for treatment of septic ankle arthritis and osteomyelitis consist of infection control, stabilization, soft tissue coverage, and the management of skeletal defects. An open approach or arthroscopic irrigation and debridement should be performed as early treatment, along with appropriate empirical antibiotics or antibiotics consistent with culture results. Serial aspiration is required if the surgery cannot be performed [3,4,8,11,14]. Inadequate treatment can cause irreversible damage to the cartilage or bone in the joint; infective synovitis; or osteomyelitis, resulting in deformity [11,15,16], sequelae, and pain and joint dysfunction. The infection may spread nearby and can cause sepsis [4,8,11].

The mortality caused by a septic joint is approximately 11%, and one-third of affected patients experience permanent joint damage [6,17]. However, the diagnosis is not always clear, and imaging tests such as hybrid 99mTc white blood cell (WBC) single-photon emission computed tomography (SPECT)/CT are often performed as part of the diagnostic task [18]. An aggressive surgical approach is required when a joint infection causes severe limb-threatening arthritis. Arthrodesis of the septic arthritic ankle joint combined with eradication of all infected tissue is the final alternative to amputation [7,19,20,21]. Many arthrodesis methods are available, but they are difficult for surgeons to perform while maintaining contralateral limb length.

Here, we describe a patient who underwent successful treatment of a secondary ankle joint deformity caused by septic ankle arthritis sequelae during staged surgery.

## 2. Case Presentation

This case report was approved by the Institutional Review Board of Soonchunhyang University Hospital (IRB No. 2021-04-017). The patient provided written informed consent for the publication of this report and the accompanying images.

A 58-year-old woman visited our clinic with increasing pain in the right ankle, which had been present for the previous 2 months. She complained of discomfort in daily life due to deformity of the ankle; limping; and severe pain in the ankle even after walking a little. The patient reported a history of right ankle injury while exiting a bus in her early 20s. She reported no other notable history regarding the right ankle. She had received intermittent treatment with injections or medicines at a local clinic for pain relief. A physical examination revealed swelling and varus deformity of the ankle joint. No redness or heat was observed on palpation (Figure 1). No tenderness or pain in the range of motion of the subtalar joint was detected. We suspected Charcot arthropathy considering the patient’s history and the severity of her ankle deformity; however, she had no other specific health issues, such as diabetes mellitus, leprosy, syphilis, chronic alcoholism, or neuropathic disease. The patient reported a visual analog scale pain score of 8 points over the previous 2 months. 

Plain radiographs of the right ankle joint revealed that the medial malleolus was nearly absent in the right ankle joint on the anteroposterior view, and severe varus deformity was observed with osteoarthritic changes because of joint space destruction. The tibiotalar joint was destroyed in the lateral view, and the tibia was subluxed forward to the talus (Figure 2). A leg length discrepancy of approximately 3 cm was confirmed by a scanogram. A three-dimensional computed tomography (CT) scan also revealed severe destruction of the tibiotalar joint and varus deformation (Figure 3). Magnetic resonance imaging showed diffuse synovial thickening of the destroyed tibiotalar joint with joint effusion (Figure 4). Increased uptake along with the soft tissue around the ankle joint was detected on hybrid 99mTc white blood cell (WBC) single-photon emission computed tomography (SPECT)/CT, but uptake was generally low in the talocrural and subtalar joints (Figure 5). The complete blood count, erythrocyte sedimentation rate, and C-reactive protein level were normal, and there were no other specific clinical findings. Based on these results, we diagnosed the patient with sequelae of septic ankle arthritis. 

A two-stage operation was planned to remove the infected lesions and correct the deformity, thus enabling limb salvage. The anterior midline approach was used for the arthrotomy and to correct the deformity. First, an open arthrotomy of the ankle joint was performed with extensive debridement of the infected tissue. The medial malleolar part of the distal tibia was almost destroyed, and the tibiotalar joint was deformed like a ball and socket joint. Because the infection involved bone beyond the articular cartilage, minimal osteotomy of the distal tibia and talus was performed (Figure 6a); an antibiotic-impregnated polymethylmethacrylate spacer was then inserted to control the infection and maintain leg length for subsequent deformity correction. Antibiotic-impregnated cement (Simplex; Stryker, Rutherford, NJ, USA; 40 g per pack, erythromycin 0.5 g per pack) was used, and 2 g of vancomycin was mixed with 40 g of cement. The ankle joint was stabilized with a delta-frame external fixator (Figure 6b,c). No bacteria were detected in a culture of infective tissues removed from the patient, and the acid-fast bacilli stain findings were negative. The patient was hospitalized and administered intravenous cefazolin for 6 weeks as antibiotic therapy. The second step of treatment comprised internal plate fixation tibiotalar arthrodesis. The original midline incision was utilized to access the surgical site, and the vancomycin spacer was removed. Frozen biopsies of surrounding tissues were sent to the Department of Pathology to determine whether the infection had cleared. Fewer than 1–2 polymorphonuclear leukocytes per high power field were observed on intraoperative frozen sections. Bony surfaces of the tibia and talus were prepared by curettage, drilling, and burring; these comprise standard methods for arthrodesis. To maintain leg length while providing structural support and stability, three cortical bone blocks were harvested from the ipsilateral ileum and then used to construct auto and cancellous chip bone grafts. The iliac crest was reconstructed with bone cement on the donor side ileum because of donor site pain [14]. The foot was fixed in a neutral position using an anterior fusion plate (Arthrex Inc., Naples, FL, USA) (Figure 7). 

The ankle joint was immobilized for 6 weeks postoperatively in a short-leg cast, with the ankle and metacarpophalangeal joint in a neutral position. A split-thickness skin graft procedure was performed 3 weeks after arthrodesis because of the need for wound dehiscence at 2 weeks after arthrodesis. The cast was changed to a Walker boot brace at 6 weeks postoperatively. Partial weight-bearing was allowed, and the range of foot motion was allowed. Full weight-bearing began at 8 weeks, and the walking boot brace was maintained for 1 additional month. Complete fusion was confirmed on plain radiographs 5 months after arthrodesis (Figure 8).

The patient was nearly asymptomatic at the 6-month follow-up, with no discomfort in her daily life and nearly normal ability to carry out full functional activities. The leg length discrepancy was corrected to 1 cm on a follow-up scanogram (Figure 9). She had no complications or recurrent symptoms at the 1-year follow-up. The American Orthopedic Foot and Ankle Society ankle-hindfoot score improved from 15 points preoperatively to 87 points postoperatively; the visual analog scale score for pain improved from 8 to 1.

## 3. Discussion

Arthrodesis is a reasonable option to prevent amputation of an infected tibiofibular joint. Various fixation methods and arthrodesis results of septic ankles have been described in the literature [7,19,20,21,22,23,24,25,26]. Representative methods include external fixation (Ilizarov, hybrid, or monofixator) [19,23,24,25,26], internal fixation [7,21], and a combination of external and internal fixation [20,22]. Thordarson et al. [26] reported good results in four patients who underwent two-stage fusion using an external monofixator. They indicated that aggressive and standardized treatment was required to salvage a septic ankle. Richer et al. [19] reported that the use of septic ankle joint arthrodesis with an external AO frame fixator is a useful method to achieve union. The union rate of primary arthrodesis is 62%, although only 39% of patients develop union after revision. Their study showed frequent complications from wound healing problems (22%), non-union (15%), pin-track infections (18%), and revisions (23%), as well as some risk factors for complications. Several studies have reported satisfactory results after arthrodesis of infected ankles with internal fixation. Klouche et al. [7] published the results of “one-stage” arthrodesis with internal fixation consisting of screw fixation, staple fixation, or a combination of both. They reported fusion in 89.5% of patients and an eradicated infection in 85.0% of patients. Simoni et al. [21] reported the results of a “two-stage” surgical treatment for arthrodesis: the first stage was accurate debridement and control of the residual space until the infection normalized, followed by ankle joint arthrodesis. The rate of union in 57 patients was 91.25%. The most frequent complications were weight-bearing ankle foot pain (27%) and ankle foot pain and surgical wound dehiscence (12.25%). Richer et al. [22] reported arthrodesis in 45 patients using internal fixation (compression screw and anterior plate, or compression screw only) and hybrid external fixation; the union rate was 86.6%. They argued that internal osteosynthesis and external fixation are reasonable from a biomechanical perspective. The rationale is that an external fixator protects against torsional rotation, but stability is weak for plantar flexion-dorsiflexion movements in the fusion gap; therefore, internal fixation is needed for neutralization. Persaud et al. [20] described a staged procedure using external and internal fixation along with infection control and an autologous pillar graft for limb salvage in a patient who had end-stage degenerative joint disease following septic ankle arthritis. Similar to the above findings, we encountered loss of limb length because of septic arthritis in our patient. We did not perform fibula osteotomy because the presence of the fibula would provide more stability during a leg length correction and fusion. Instead, we constructed pillars from two hard tricortical bones in the iliac bone and then inserted and fixed these pillars between the defects. Because the patient did not report pain in the subtalar joint and no arthritic changes were observed via imaging, we performed tibiotalar arthrodesis using anterior plating rather than tibiotalocalcaneal fusion arthrodesis using intramedullary nails.

## 4. Conclusions

We have described a rare case of a staged limb salvage procedure in a patient with chronic septic arthritis sequelae. Arthrodesis was the final alternative to avoid amputation, considering the patient’s overall condition. Our findings indicate that an aggressive surgical approach is required for arthrodesis when joint infection causes severe limb-threatening arthritis. Moreover, for patients with severe joint deformity because of septic ankle sequelae, staged arthrodesis is a reliable method to remove infected lesions, solve soft tissue problems, correct deformities, and maintain leg length.

## Figures and Tables

**Figure 1 ijerph-18-12473-f001:**
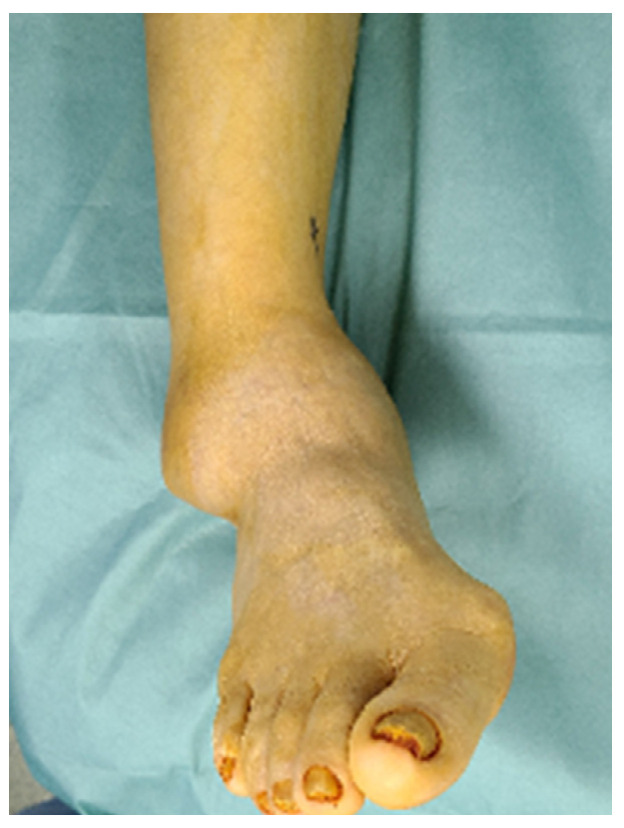
Gross photographs showing varus deformity of the ankle joint with swelling.

**Figure 2 ijerph-18-12473-f002:**
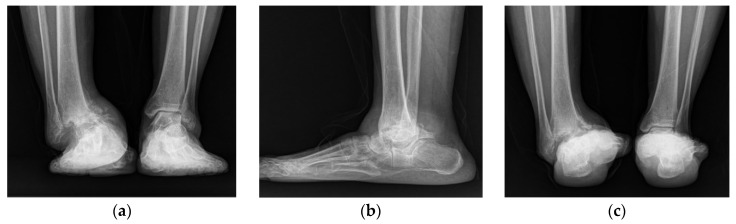
Plain anteroposterior (**a**), lateral (**b**), and heel alignment (**c**) radiographs showing an ankle with severe varus deformity, and osteoarthritic changes with destruction of the tibiotalar joint.

**Figure 3 ijerph-18-12473-f003:**
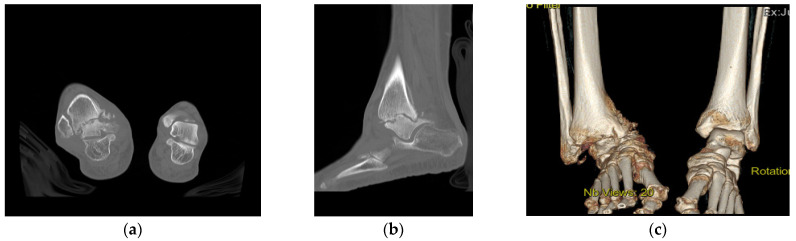
Preoperative coronal CT image showing a severe defect in the distal tibia and nearly undetectable medial malleolus (**a**). Sagittal CT scan showing a destroyed tibiotalar joint and anterior tibia migration to the talus (**b**). The 3D reconstructed CT image shows severe varus deformity of the right ankle joint (**c**).

**Figure 4 ijerph-18-12473-f004:**
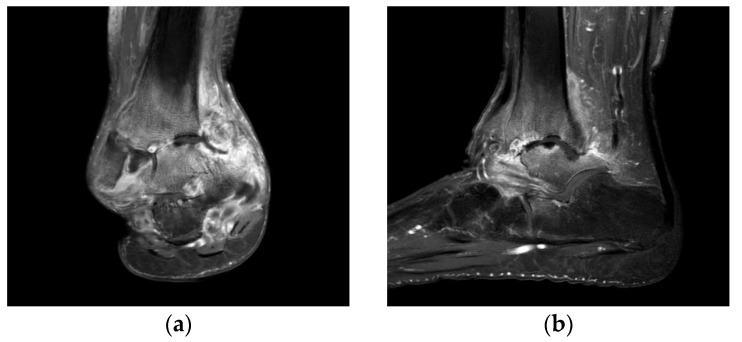
Preoperative T2-weighted coronal (**a**) and sagittal (**b**) magnetic resonance images showing a septic ankle joint with diffuse synovial thickening, increased joint effusion with surrounding soft tissue swelling of the distal tibia, and talus osteomyelitis.

**Figure 5 ijerph-18-12473-f005:**
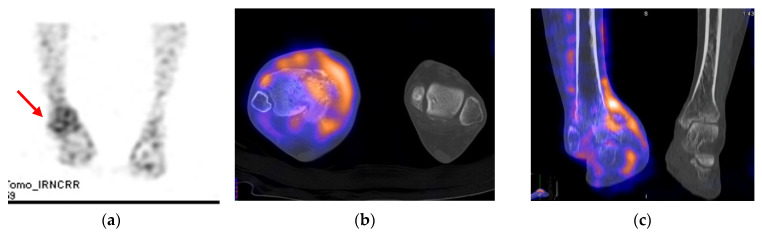
Hybrid 99mTc WBC SPECT/CT images showing increased uptake along the soft tissue around the ankle joint, as well as generally low uptake in the talocrural and subtalar joints. 99m Tc WBC SPECT (**a**)**,** hybrid WBC SPECT/CT axial image (**b**), and coronal image (**c**).

**Figure 6 ijerph-18-12473-f006:**
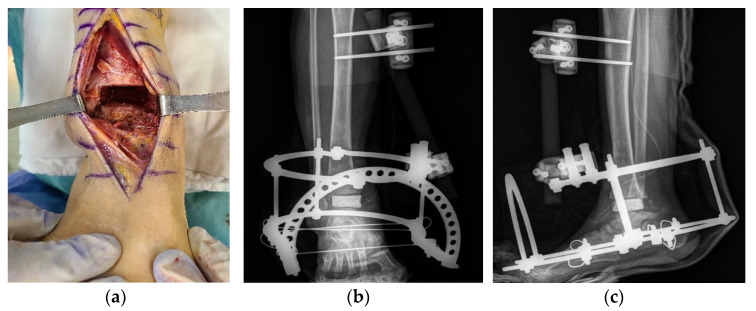
Intraoperative gross photograph showing open arthrotomy of the ankle joint with extensive debridement of infected tissue (**a**). Plain anteroposterior (**b**) and lateral radiographs (**c**) taken after the first stage showing insertion and fixation of a vancomycin polymethylmethacrylate spacer with an external fixator.

**Figure 7 ijerph-18-12473-f007:**
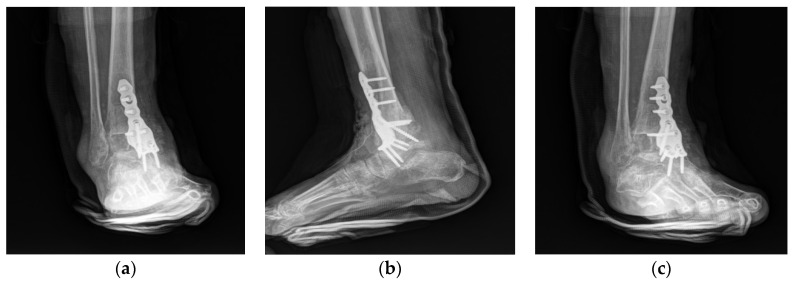
Postoperative plain anteroposterior (**a**)**,** lateral (**b**), and oblique (**c**) radiographs showing deformity correction and anterior plating with a bone graft.

**Figure 8 ijerph-18-12473-f008:**
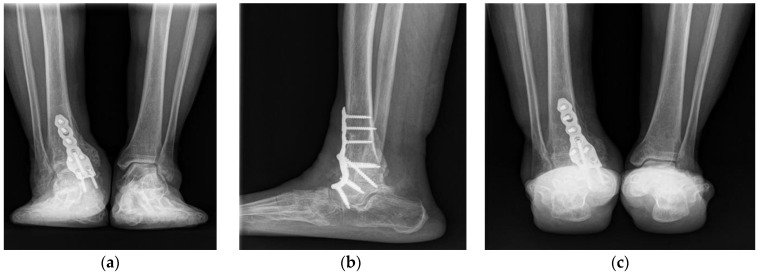
Postoperative plain anteroposterior (**a**), lateral (**b**), and heel alignment (**c**) radiographs at the 5-month follow-up examination showing complete fusion and corrected alignment.

**Figure 9 ijerph-18-12473-f009:**
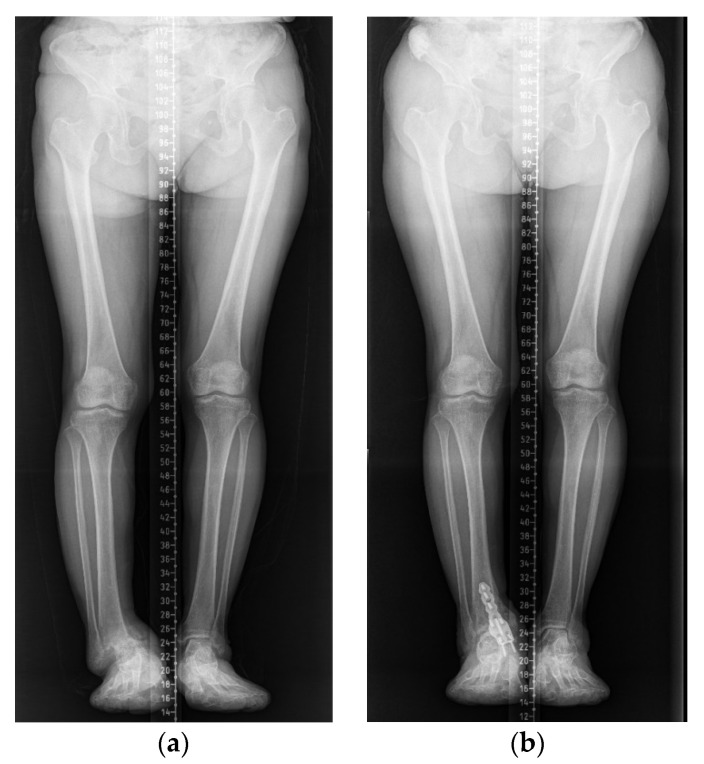
Scanograms showing leg length discrepancy correction from 3 cm before surgery (**a**) to 1 cm after surgery (**b**).

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
