# Peer review of "Staged Joint Arthrodesis in the Treatment of Severe Septic Ankle Arthritis Sequelae: A Case Report"

_ijerph, 2021, doi:10.3390/ijerph182312473_

Round 1

Reviewer 1 Report

Dear authors. This report has a nice and well presented review; because the case has interesting imaging tools, it will be useful if you consider  to add some sentences to the introduction regarding to the utility of the different imaging probes in the diagnostic approach in this clinical setting.

Author Response

Response to Reviewer 1 Comments

First of all, thank you for the good review.

Comments and Suggestions for Authors

Dear authors. This report has a nice and well presented review; because the case has interesting imaging tools, it will be useful if you consider  to add some sentences to the introduction regarding to the utility of the different imaging probes in the diagnostic approach in this clinical setting.

Response 1:

Finding the causes of secondary OA is very difficult. In particular, WBC SPECT can be helpful among various imaging tools for diagnosing septic arthritis.

As reviewer 1 pointed out, we mentioned in the introduction WBC SPECT/CT, which is used to make a definitive diagnosis of joint infection.

Thank you!!

The authors would like to thank the reviewer once again for the excellent advice.

Reviewer 2 Report

Well Written and well presented. Just a few suggestions:

  • The introduction section needs to be expanded. I believe that the discussion between lines 172-202 is more appropriate in the Introduction section since it defines the problem and provides background on the technique you used. 
  • In the Introduction section, could you provide more instances of why ankle sepsis may occur?
  • Line 79 - Could you make it clear what the injections and treatments were meant to do? Clear infection? Reduce pain?
  • Line 164 - What were the mobility issues before the surgery so that we can compare?

Author Response

Response to Reviewer 2 Comments

First of all, thank you for the good review.

Comments and Suggestions for Authors

Well Written and well presented. Just a few suggestions:

The introduction section needs to be expanded. I believe that the discussion between lines 172-202 is more appropriate in the Introduction section since it defines the problem and provides background on the technique you used.

Response 1:

As pointed out, in the introduction part, the contents in the discussion have been changed and adapted and attached. Thank you.

In the Introduction section, could you provide more instances of why ankle sepsis may occur?

Response 2:

As you pointed out, the introduction part describes the cause of ankle sepsis.

Thank you.

Line 79 - Could you make it clear what the injections and treatments were meant to do? Clear infection? Reduce pain?

Response 3:

As you pointed out, I added that the patient was prescribed an "analgesic" and not an "antibiotic".

Line 164 - What were the mobility issues before the surgery so that we can compare?

Response 4:

As pointed out, what the patient complained about when he first came to the hospital was described in the case description. Thanks for the accurate point.

Thank you!!

The authors would like to thank the reviewer once again for the excellent advice.

Reviewer 3 Report

Originality:  This is a very short paper, providing a minimum of information about a case study, and constitutes a very minor contribution to the literature.  The introduction section did not provide a clear rationale for carrying out the study (for example, why is your research question important? What gap in the literature is the study addressing?). I suggest to describe in this section only with the information related with the state of art related with ankle injury, staged joint arthrodesis and treatment of severe septic ankle arthritis sequelae.
Thus, I suggest in the introduction section should be improved, with more details about prevalence, see research of Kazemi et al https://pubmed.ncbi.nlm.nih.gov/28843163/ related with rehabilitative for subjects with and without functional ankle instability. Furthemore, to revise the research of Casado-Hernández  et al  related with the  Association between anterior talofibular ligament injury and ankle tendon, ligament, and joint conditions revealed by magnetic resonance imaging https://pubmed.ncbi.nlm.nih.gov/33392013/

Methodologically Sound:  As a case study report it is rather hard to go wrong methodologically, and the paper conforms to the standard.

Follows Appropriate Ethical Guidelines: Whilst there is no obvious declaration of ethical approval, it would appear to be a report of actions taken as part of normal clinical practice (as a case study report), and thus is acceptable. 

Has results which are clearly presented and support the conclusions: Again, it conforms to the usual format for the presentation of a case study, although the content is very sparce.  It is, however, appropriate enough, and does report a rare case likely to be of interest to a healthcare audience.  

Overall Scientific Quality:  As a minor case study report it lacks scientific depth, but effectovely is intended only to report the occurence of a typical case and to highlight the importance of correct disgnosis, and on these grounds merits attention. 

Correctly References Previous Relevant Work:  It not appears to reference prior work succinctly and accurately. 

Importance/Interest: Although marked by its brevity, the content is of interest, particularly to clinicians such of Orthopaedic Surgery who work in this area and physiotherapist  who may need to be aware of the variant forms of this illnes.  

Author Response

Response to Reviewer 3 Comments

First of all, thank you for the good review.

Comments and Suggestions for Authors

Originality:  This is a very short paper, providing a minimum of information about a case study, and constitutes a very minor contribution to the literature.  The introduction section did not provide a clear rationale for carrying out the study (for example, why is your research question important? What gap in the literature is the study addressing?). I suggest to describe in this section only with the information related with the state of art related with ankle injury, staged joint arthrodesis and treatment of severe septic ankle arthritis sequelae.
Thus, I suggest in the introduction section should be improved, with more details about prevalence, see research of Kazemi et al https://pubmed.ncbi.nlm.nih.gov/28843163/ related with rehabilitative for subjects with and without functional ankle instability. Furthemore, to revise the research of Casado-Hernández  et al  related with the  Association between anterior talofibular ligament injury and ankle tendon, ligament, and joint conditions revealed by magnetic resonance imaging https://pubmed.ncbi.nlm.nih.gov/33392013/

Response 1:

I have edited the introduction as you pointed out. As you advised, referring to studies by Kazemi et al. and Casado-Hernández et al., the mechanism by which discomfort can occur when a patient develops instability with septic arthritis sequelae was suggested. However, in revising the introduction, there was a procedure to supplement the contents of the discussion with an introduction according to reviewer 2's comment.

Thank you for making the paper more meaningful.

The authors would like to thank the reviewer once again for the excellent advice.

Round 2

Reviewer 3 Report

The authors have Previous requests, not sufficiently addressed. In the references section, Where are the references you claimed to have added? Casado et al? Kazemi  et al.? 

Author Response

Sorry. There was a problem in the process of uploading the revised manuscript. The authors have uploaded the revised manuuscript again.

It is listed in reference 12,13.

Thank you for making the paper more meaningful.

The authors would like to thank the reviewer once again for the excellent advice.
